# CONTINUUM: Restoring the Contiguous Tensor Abstraction Efficiently for Dynamic AI Workloads via Hardware Virtualization

**Yangyu Zhang** [* 1 2] **Shuoming Zhang** [* 1 2] **Chunwei Xia** [3] **Shuaijiang Li** [1 2] **Zhicheng Li** [1 2] **Ruiyuan Xu** [1 2]
**Zheming Yang** [1 2] **Lei Chen** [2] **Yuan Wen** [4] **Guangli Li** [1 2] **Xiaobing Feng** [1 2] **Huimin Cui** [1 2 5] **Jiacheng Zhao** [1 2]

## Abstract

Emerging LLM workloads demand extreme memory agility. However, state-of-the-art inference systems (e.g., vLLM) rely on software-defined paging, which sacrifices the contiguous tensor abstraction. This rigid interface exposes fragmentation complexity to developers, imposing a severe engineering burden that stifles algorithmic innovation. We introduce CONTINUUM, a tensor memory virtualization subsystem implemented as a PyTorch extension. By bypassing serialized OS bottlenecks through a lightweight GPU driver extension, CONTINUUM significantly reduces mapping costs by orders of magnitude, from milliseconds to microseconds. Built atop this low-latency API, CONTINUUM provides *Elastic Tensor*, a set of flexible tensor operations that natively support complex memory dynamics and zero-copy topological aliasing. Evaluations demonstrate that CONTINUUM achieves significantly higher throughput across diverse dynamic scenarios, effectively lowering the barrier to implementing next-generation LLM applications.

## 1. Introduction

The landscape of Large Language Models (LLMs) is shifting from simple chat completions (OpenAI; Google) to sophisticated, dynamic workflows (Chase; Anysphere, Inc.; anthropics). Cutting-edge research now focuses on **Complex Reasoning Topologies** (e.g., Tree-of-Thoughts (Yao et al., 2023)), **Advanced Caching Strategies** (e.g., Radix Caching (Zheng et al., 2024), Position-Independent

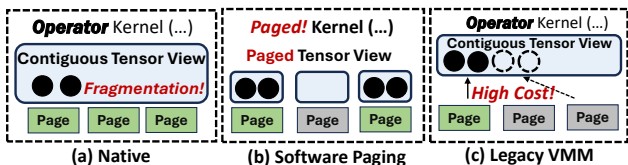

*Figure 1.* **Memory Management Paradigms.** (a) **Native:** Suffers from severe fragmentation. (b) **Software Paging:** Mitigates fragmentation via software paging but sacrifices kernel transparency and resource flexibility. (c) **Legacy VMM:** Decouples allocation via standard hardware APIs but incurs high mapping cost.

Caching (Hu et al., 2025)), and **Dynamic Model Compositions** (e.g., Multi-LoRA (Chen et al., 2024; Sheng et al., 2024), Engram $N$-grams (Cheng et al., 2026)). These emerging workloads demand extreme algorithmic agility: tensors should grow indefinitely, fork efficiently, and share memory context flexibly across requests.

However, a fundamental gap exists between these model requirements and current system capabilities. Deep learning frameworks like PyTorch (Paszke et al., 2019) rely on the *Contiguous Tensor Abstraction*, enabling high-performance and easy-to-use linear addressing but suffering from severe fragmentation under dynamic serving loads. To avoid the overhead of cudaMalloc, these frameworks utilize monolithic user-space caching allocators, which exacerbate internal fragmentation and often drop effective utilization to as low as 20% (Kwon et al., 2023).

To address fragmentation, state-of-the-art serving systems (e.g., vLLM (Kwon et al., 2023) and SGLang (Zheng et al., 2024)) adopt software-defined paging via the *Paged Tensor Abstraction*. While effective for specific structures like KV caches, this approach introduces a rigid segregation between paged and contiguous memory pools. Standard tensors requiring contiguous layouts (e.g., weights or intermediate activations) cannot utilize fragmented blocks within the paged pool, leading to resource underutilization. Furthermore, this design tightly couples memory management with kernel implementation, imposing a severe programming and optimization burden. Since the non-contiguous address space invalidates vendor-optimized libraries (e.g., cuBLAS (NVIDIA, 2024)), developers are forced to refactor

---

*Equal contribution [1] State Key Lab of Processors, Institute of Computing Technology, Chinese Academy of Sciences [2] University of Chinese Academy of Sciences [3] University of Leeds [4] University of Aberdeen [5] XCORESIGMA CO.,LTD. Correspondence to: Jiacheng Zhao <zhaojiacheng@ict.ac.cn>.

*Proceedings of the 43rd International Conference on Machine Learning*, Seoul, South Korea. PMLR 306, 2026. Copyright 2026 by the author(s).

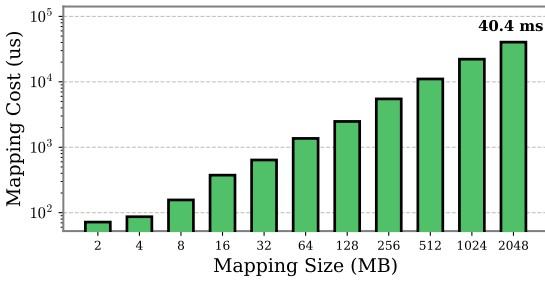

*Figure 2.* The control-plane overhead of mapping memory via standard driver APIs scales linearly with allocation size. Mapping gigabyte-sized regions incurs prohibitive millisecond-level latency.

algorithmic logic into complex *paged-aware* kernels (e.g., for S-LoRA (Sheng et al., 2024)). This process is not only engineering-intensive, often lagging behind non-paged primitives by 20–30% (Prabhu et al., 2025), but also hinders the deployment of advanced aliasing patterns which are difficult to implement efficiently atop rigid software block tables.

*Major Insight.* To resolve the tension between memory efficiency and programming agility, we adopt a principle of *strict decoupling between logical tensor views and physical memory residence*. Unlike existing framework abstractions that rely on complex software-level striding over pre-allocated buffers, we identify that the GPU's native Memory Management Unit (MMU) offers a powerful, yet underutilized, mechanism for dynamic tensor virtualization. Our core insight is to leverage this hardware capability to implement tensors with the attribute of *eager virtual reservation with on-demand physical mapping*. By maintaining a contiguous virtual address space for upper-layer kernels while transparently managing fragmented physical pages in the background, this strategy renders physical memory fragmentation invisible to the application. This fulfills the primary goal of modern frameworks: enabling unmodified, vendor-optimized kernels to operate efficiently on rapidly evolving architectures without bespoke programming or data movement.

*Challenge.* In principle, standard CUDA Virtual Memory Management (VMM) APIs (Perry & Sakharnykh) should enable this capability. However, applying VMM to fine-grained, latency-sensitive inference is blocked by *prohibitive control-plane latency*. Designed for coarse-grained resource management, legacy driver APIs (e.g., `cuMemMap`) rely on serialized, lock-heavy control paths. As quantified in Figure 2, the mapping cost scales linearly with allocation size, escalating from 52.9 $\mu$s for a single 2MB page to 40.4 ms for a 2GB range. This millisecond-scale overhead is orders of magnitude higher than a typical decoding step, which makes standard VMM APIs impractical for the high-frequency allocation demands of interactive serving.

*Takeaway.* We argue that overcoming these limitations requires rethinking how infrastructure exposes hardware capabilities to the runtime. Consequently, our core contribution extends beyond the mere adoption of VMM; rather, it lies in rendering GPU VMM practical on the latency-sensitive serving critical path. Specifically, we modify the GPU driver to transform the VMM control plane into a transactional, batched remapping primitive, thereby reducing mapping costs to the microsecond scale. To this end, we present CONTINUUM, a tensor virtualization middleware designed as a flexible extension for deep learning frameworks. With the latency bottleneck removed, complex algorithmic patterns can be re-implemented via lightweight virtual-to-physical remapping rather than cumbersome engineering. The same primitive supports both model-level storage dynamics (elastic KV caches, multi-LoRA, Engram $N$-grams) and sequence-level topological aliasing (zero-copy branching for Tree-of-Thoughts, context caching). By actively reshaping the virtual address space to match these algorithmic needs, CONTINUUM enables unmodified, vendor-optimized kernels to operate on flexible memory layouts with zero-copy remapping.

CONTINUUM bridges the gap between algorithmic agility and system efficiency through three contributions:

- **A high-performance VMM backend:** We identify that standard driver latency is the primary bottleneck preventing hardware-level virtualization. We implement a high-performance VMM backend that replaces serialized OS operations with *batched, transactional hardware updates*, reducing mapping costs to enable microsecond-scale responsiveness.
- **Tensor Virtualization Middleware:** Built atop this backend, we propose CONTINUUM, a transparent extension that restores the contiguous tensor abstraction. By unifying Model-Level storage elasticity (e.g., KV cache, LoRA slot and $N$-gram embedding) and Sequence-Level topological aliasing (e.g., Tree-of-Thought, Prefix Caching and Position-Independent Caching) under a single *Elastic Tensor* interface, it simplifies the implementation of complex dynamic workloads without bespoke kernel programming.
- **Extensive Evaluation:** We evaluate CONTINUUM across diverse dynamic scenarios, spanning from long-context generation to emerging agentic paradigms. Results demonstrate that CONTINUUM achieves throughput comparable to or significantly surpassing state-of-the-art systems, while maintaining compatibility with unmodified, vendor-optimized kernels.

## 2. Background & Related Work

Current LLM serving infrastructures face a dichotomy: they need to support increasingly *dynamic model structures*

(Model-Level) while managing complex, non-linear *reasoning topologies* (Sequence-Level). Existing solutions, however, largely rely on static or software-defined abstractions that lack the necessary elasticity.

## 2.1. Model-Level Storage Dynamics

Modern workloads require tensors that are physically elastic, capable of expanding, shrinking, or becoming sparse on demand without breaking logical contiguity.

**Elastic KV Caches and Dynamic Growth.** The fundamental bottleneck in LLM serving is the Key-Value (KV) cache, which grows linearly with sequence length. In scenarios like infinite-context generation or Retrieval-Augmented Generation (RAG) (Lewis et al., 2020), activation tensors fluctuate wildly in size. Current systems (e.g., vLLM (Kwon et al., 2023)) manage this via PagedAttention, which pre-allocates fixed-size blocks. However, this software-defined approach lacks real-time *physical* elasticity. It cannot dynamically resize or compact memory pool without explicit data movement, limiting the system's ability to handle bursty, high-variance batch sizes efficiently.

**Dynamic and Sparse Model Components.** Emerging workloads increasingly rely on dynamic memory footprints, ranging from the on-the-fly composition of variable-rank LoRA adapters (Hu et al., 2022) (e.g., for code (Chang et al., 2024; Zhang et al., 2024) or style (Shah et al., 2024; Po et al., 2024; Xu et al., 2025)) to the irregular access patterns of sparse $N$-gram embeddings in Engram (Cheng et al., 2026). While systems like S-LoRA (Sheng et al., 2024) and Punica (Chen et al., 2024) manage these heterogeneous components via non-contiguous memory pools, they inherently require specialized "scatter-gather" kernels to stitch fragmented physical blocks. This breaks compatibility with vendor-optimized libraries (e.g., cuBLAS) and imposes a severe engineering burden to bridge the gap between logical algorithmic intent and rigid physical block tables.

## 2.2. Sequence-Level Topological Dynamics

Beyond storage, reasoning workflows demand topological agility, the ability to fork or deduplicate context states instantly.

**Reasoning Topologies.** Complex reasoning tasks employ non-linear exploration strategies like Beam Search (Shu & Nakayama, 2018) and Tree-of-Thoughts (ToT) (Yao et al., 2023). These algorithms generate a branching topology where a single parent state forks into multiple hypotheses. Supporting this in software-paged systems is inefficient: creating a new branch often involves deep-copying block tables or managing complex reference counts in user-space. This "logical branching" overhead scales poorly with tree depth and width, becoming a bottleneck for high-throughput

agentic reasoning.

**Prefix and Position-Independent Caching.** Eliminating redundant computation is critical for throughput. *Prefix Caching* (e.g., in SGLang (Zheng et al., 2024)) allows requests to share common prompt prefixes via a Radix Tree. More recently, *Position-Independent Caching (PIC)* (Yao et al., 2025; Hu et al., 2025) attempts to reuse context blocks regardless of their absolute position in the sequence. Implementing PIC on rigid block tables is notoriously difficult because standard Attention mechanisms bind physical blocks to specific logical indices. Achieving flexible, position-agnostic reuse requires a virtualization layer that can alias memory logically without moving data physically.

## 2.3. Why a New VMM Primitive

Hardware memory virtualization offers a natural solution: by leveraging the GPU MMU, runtimes can transparently support elasticity and aliasing while preserving contiguous tensor views for standard kernels. Recent systems validate this approach: GMLake (Guo et al., 2024) stitches fragmented training memory, and vAttention (Prabhu et al., 2025) backs KV caches via CUDA VMM. However, they still rely on the *unmodified vendor VMM control plane*. Millisecond-scale, size-linear control-plane latency makes fine-grained remapping prohibitively slow. Consequently, prior works can only *hide or amortize* this overhead through overlap scheduling, pre-mapping, or coarse-grained management, rather than eliminating the root bottleneck. These mitigations inevitably break down when remapping lies on the critical path and cannot be overlapped. Furthermore, these point solutions fail to restore a unified contiguous abstraction for both model- and sequence-level dynamic memory management.

A fundamentally new VMM primitive is therefore needed to (i) accelerate the control plane, making fine-grained remapping viable on the critical path, and (ii) expose a general tensor-level abstraction rather than workload-specific optimizations.

## 3. System Design

We design CONTINUUM as a transparent tensor virtualization middleware interposed between the deep learning framework (e.g., PyTorch) and hardware-level memory primitives. Unlike monolithic inference systems that enforce rigid memory management policies, CONTINUUM operates as a flexible *runtime extension*. Its primary function is to virtualize the GPU's hardware capabilities, presenting a coherent, contiguous memory view to upper-layer applications while efficiently orchestrating fragmented physical pages in the backend.

The core design philosophy of CONTINUUM is *decoupling*

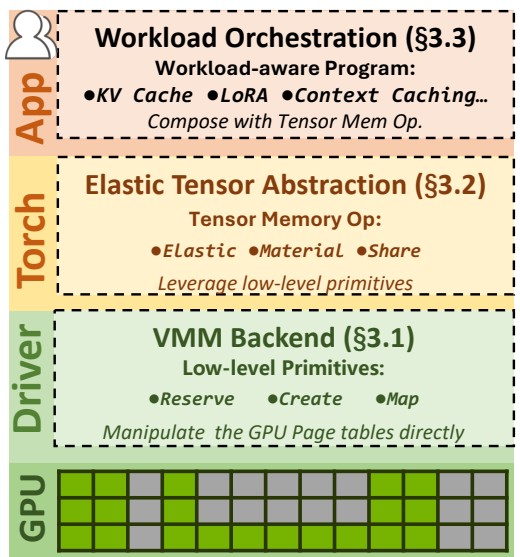

*Figure 3.* Overview of CONTINUUM.

*Table 1.* Comparison of standard CUDA VMM against CONTINUUM primitives. CONTINUUM reduces overhead by replacing serialized driver logic with hardware-instantiated metadata.

| Standard CUDA VMM | CONTINUUM VMM |
|---|---|
| `cuMemAddressReserve`
*Logical bookkeeping only* | `MemAddressReserve`
***Eagerly creates** page tables* |
| `cuMemCreate`
*Allocates raw physical pages* | `MemCreate`
*Allocates pages &*
***pre-computes PTEs*** |
| `cuMemMap` / `SetAccess`
*Serialized, lock-heavy calls* | `MemMap`
***Batched, transactional** PTE*
*population* |

*of the logical tensor view from physical memory residence.* By bridging the semantic gap between standard framework operators and fragmented hardware resources, CONTINUUM restores the *Contiguous Tensor Abstraction* essential for algorithmic agility.

As illustrated in Figure 3, CONTINUUM comprises three hierarchical layers:

- **Enhanced VMM Engine (Backend Primitive):** A high-performance driver extension that exposes the GPU MMU as a microsecond-scale programmable resource. It replaces serialized OS operations with batched, transactional updates to eliminate the mapping latency barrier.
- **Elastic Tensor Abstraction (Frontend Interface):** A PyTorch extension that decouples logical views from physical storage. It allows unmodified kernels to operate on contiguous virtual addresses backed by fragmented, on-demand physical memory.
- **Workload Orchestration (Policies):** A workload policy layer that translates specific algorithmic intent into physical memory operations. It implements specialized strategies for structural elasticity (e.g., dynamic KV growth) and topological aliasing (e.g., zero-copy branching) to natively support diverse dynamic workloads.

### 3.1. The VMM Backend

Standard VMM APIs suffer from prohibitive latency at fine page granularity due to serialized control-plane overheads. Specifically, each mapping operation incurs the cumulative cost of context switches, driver-level lock contention, and synchronous PCIe round-trips. This latency often eclipses the computation time of a single decoding step, rendering

dynamic hardware-level memory management impractical for fine-grained inference.

To overcome this, CONTINUUM interacts directly with the GPU's Memory Management Unit (MMU) by modifying the NVIDIA GPU open source driver (NVIDIA). Its core philosophy is the *decoupling of metadata preparation from mapping commitment*. By separating the heavyweight management of page table structures (Control Plane) from the lightweight application of mappings (Data Plane), CONTINUUM removes *Page Table Entry* (PTE) construction from the critical path. The resulting transactional interface (summarized in Table 1) maintains semantic alignment with standard CUDA VMM APIs primitives while enabling hardware-accelerated performance.

**Eager Virtual Reservation.** Unlike standard APIs that perform lazy logical bookkeeping, `MemAddressReserve` eagerly instantiates the complete page table hierarchy (from PML4 down to leaf Page Directories) directly in GPU memory. By physically allocating these physical backing pages upfront and initializing their point-to structures, CONTINUUM ensures the address translation skeleton is strictly hardware-compliant and physically resident. This guarantees that the MMU traversal path is established before any inference request arrives, leaving only the leaf PTEs to be populated.

**Offline PTE Pre-calculation.** CONTINUUM leverages the insight that *tensor page attributes are static*. Instead of computing hardware bits during the critical path, `MemCreate` performs bitwise pre-calculation in host memory. When physical pages are allocated, CONTINUUM immediately calculates their hardware-specific PTE binary formats and stores them in a *Host PTE Buffer*. This transforms the complex logic of *mapping* into a simplified bitstream preparation process, completely decoupled from the GPU execution timeline.

**Batched Transactional Mapping.** With the GPU page tables structurally ready and source PTEs pre-calculated, the

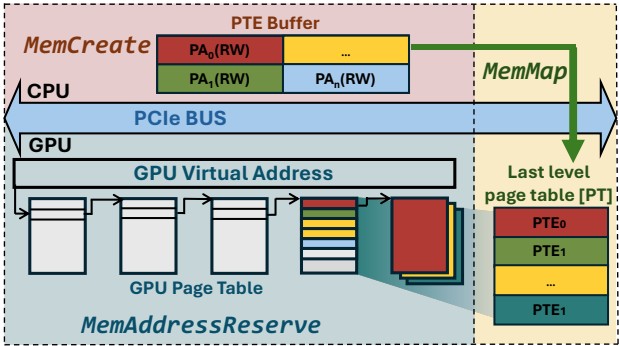

*Figure 4.* Workflow of CONTINUUM's fast VMM backend: Decoupling metadata from mapping commitment via direct MMU interaction.

mapping operation (MemMap) is reduced to a raw memory overwrite. This primitive executes a transactional update: the Host CPU initiates a vectorized high-speed copy (via PCIe BAR or DMA) to inject PTEs directly into the GPU's physical page tables. As illustrated in Figure 4, this collapses the complexity of mapping $N$ pages from $O(N)$ serialized kernel transitions to a single hardware transaction, effectively bypassing driver-level software locks. Conversely, deallocation (MemUnmap) performs a batched zero-write to invalidate target PTEs; this severs the translation immediately while preserving the underlying page table structure for rapid reuse. All page table updates are validated and committed exclusively in kernel space; user runtimes never observe physical addresses nor issue unchecked mappings.

## 3.2. Elastic Tensor Abstraction

To bridge the semantic gap between high-level deep learning frameworks and low-level VMM primitives, we introduce the *Elastic Tensor* abstraction via Python-native bindings. To upper-layer computation kernels (e.g., FlashAttention, cuBLAS), an *Elastic Tensor* is indistinguishable from a standard contiguous Tensor. Internally, however, it decouples the virtual address space from physical residency and manages placement and sharing explicitly via programmable VMM primitives, as shown in Figure 3.

**Elastic Tensor Reservation (`elastic`).** CONTINUUM allows frameworks to reserve large, contiguous virtual address (VA) ranges without committing physical memory. For example, Tensor.elastic(shape) returns a handle to a valid VA range backed by a skeletal page table. This capability is critical for dynamic workloads with unpredictable footprints, such as KV cache growth and LoRA slot expansion. By securing the virtual range upfront, CONTINUUM guarantees that tensors remain logically contiguous as they grow, eliminating expensive data migration or re-allocation. *System Interaction.* This operation invokes MemAddress-Reserve, which eagerly instantiates the page directory

hierarchy in GPU memory, minimizing the overhead of subsequent leaf-level PTE updates.

**Physical Materialization (`materialize`).** After a virtual range is reserved, CONTINUUM incrementally binds physical pages to specified subranges of the VA space. The materialize(offset, size) operation provisions physical memory on demand and installs the corresponding page-table mappings. This design enables *just-in-time* provisioning: physical memory is committed only when data is actually generated, avoiding peak pre-allocation and internal fragmentation.
*System Interaction.* This operation triggers the transactional MemMap. By batching PTE updates for multiple pages into a single DMA transaction, CONTINUUM achieves microsecond-scale cost suitable for token-wise allocation.

**Zero-Copy (`share`).** To support efficient state sharing and branching, Elastic Tensors provide a zero-copy aliasing primitive via dst.share(src). Rather than copying data, this operation creates a new virtual view (dst) that maps to the same physical pages as the source tensor (src). This mechanism is particularly effective for autoregressive workloads, where historical states are immutable. Shared physical pages are treated as read-only prefixes, while newly generated suffix tokens are appended using freshly materialized pages, preserving isolation between branches.
*System Interaction.* Unlike OS-level Copy-on-Write, which relies on page faults and write protection, aliasing in CONTINUUM is purely metadata-driven. The runtime installs additional page-table mappings to reference existing physical pages via MemMap, enabling state duplication without data movement or fault handling.

**Deallocation (`release`).** Upon releasing an Elastic Tensor, CONTINUUM decrements the reference counts of its underlying physical handles. The associated virtual address range is immediately invalidated to prevent dangling access. However, to ensure safety in shared scenarios, physical pages are returned to the global pool only when their reference count drops to zero.
*System Interaction:* This invokes MemUnmap to batch-update target GPU PTEs to an invalid state. A TLB flush is subsequently issued to ensure architectural consistency before the physical memory is reclaimed.

## 3.3. Workload Orchestration

Leveraging the *Elastic Tensor* abstraction, CONTINUUM orchestrates memory resources in a workload-aware manner by translating high-level algorithmic intent into low-level virtual-to-physical remapping operations. Rather than hard-coding policies for specific applications, CONTINUUM composes the tensor memory operators so that diverse dynamic workloads can be expressed naturally while preserving kernel transparency. We observe that emerging LLM workloads

exhibit two orthogonal dimensions of dynamism, and we therefore organize this section around two representative scenarios that illustrate how *Elastic Tensors* are programmed in practice.

**Scenario I: Model-Level Structural Elasticity.** This scenario concerns tensors whose *physical footprint* is unknown a priori or inherently fragmented. Instead of pre-allocating contiguous buffers, CONTINUUM represents such tensors as virtual address ranges backed by physical pages that are bound on demand.

*On-Demand Growth (KV Cache).* The KV cache grows monotonically with generation length and is a central bottleneck in LLM serving. In CONTINUUM, the runtime first reserves a large virtual address range for the KV cache, without committing physical memory. As decoding progresses, physical pages are incrementally materialized only for newly generated tokens. This lazy binding avoids peak pre-allocation and eliminates buffer migration when the working set expands.

*Virtual Stitching (LoRA / Engram).* For fragmented model components such as variable-rank LoRA adapters or sparse Engram tables, CONTINUUM packs physical pages tightly in memory while exposing a contiguous virtual view to computation kernels. By stitching scattered pages into a linear virtual address range, CONTINUUM allows unmodified, vendor-optimized dense kernels (e.g., cuBLAS GEMM) to operate efficiently on irregular components without gather-style access or kernel refactoring.

**Scenario II: Sequence-Level Topological Aliasing.** This scenario concerns the *logical relationships* between tensors across requests or hypotheses. Here, CONTINUUM leverages hardware page tables to manipulate topology directly, enabling zero-copy sharing without data movement.

*Zero-Copy Branching.* For non-linear reasoning patterns such as beam search and Tree-of-Thoughts, a single sequence may fork into multiple hypotheses. Instead of copying KV states, CONTINUUM creates new logical views by aliasing the parent's physical pages. Branching thus becomes a single metadata operation, while new physical pages are allocated only for divergent suffix tokens.

*Context Deduplication.* CONTINUUM provides global context deduplication through many-to-one aliasing, enabling both position-dependent prefix reuse and position-independent reuse across arbitrary sequence positions.

**Case Study: KV Cache Management with Prefix Reuse** We illustrate workload orchestration using KV cache management as a running example (see Figure 5). CONTINUUM implements KV caching using a decoupled virtual–physical tensor abstraction that hides all mapping complexity behind a unified tensor interface. When prefix caching is enabled,

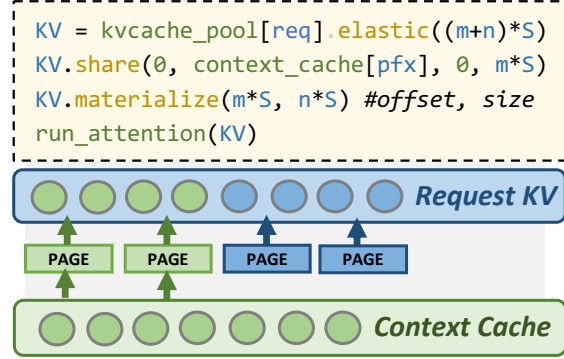

```
KV = kvcache_pool[req].elastic((m+n)*S)
KV.share(0, context_cache[pfx], 0, m*S)
KV.materialize(m*S, n*S) #offset, size
run_attention(KV)
```

*Figure 5.* CONTINUUM constructs a single contiguous KV view by reserving VA space, aliasing cached prefix pages, and materializing new suffix pages on demand.

CONTINUUM stores precomputed KV states as cached physical pages. For a request consisting of a cached prefix of length $m$ and an uncached suffix of length $n$, CONTINUUM constructs a unified KV view by reserving a contiguous virtual range of size $(m + n) \times S$, aliasing the prefix region to cached pages, and materializing new pages for the suffix. This process exposes a single contiguous tensor to upper-layer kernels while achieving zero-copy prefix reuse and elastic growth internally. Prefix caching is not a special-case optimization; rather, it arises naturally from composing the Elastic Tensor primitives. The workflow provides a zero-copy alternative to torch.cat: conventional concatenation materializes a new buffer and copies the input tensors into it, whereas CONTINUUM uses elastic and share to logically stitch tensors by remapping their underlying physical pages into a contiguous virtual range. The same mechanism also supports fine-grained, page-level sharing within a single tensor instance: a unified tensor view can be backed by a mixture of shared and private physical pages, an ownership pattern that standard frameworks cannot express because each tensor is tied to a monolithic physical allocation.

## 4. Evaluation

The evaluation answers two questions:

**Q1:** Does the CONTINUUM engine overcome the prohibitive mapping latency of the standard CUDA VMM control plane?

**Q2:** Does CONTINUUM generalize across emerging dynamic LLM workloads?

### 4.1. Experimental Setup

**Hardware & Implementation.**

We evaluate CONTINUUM on a single NVIDIA A100 SXM GPU with CUDA 12.6 and PyTorch 2.6, hosted by an Intel

Xeon Gold 6430 CPU with PCIe Gen4 interconnects providing 64 GB/s bidirectional bandwidth. Our implementation is organized into three layers with clearly defined interfaces:

- **Kernel module:** An additive extension to the NVIDIA Open Kernel Modules (NVIDIA) that exposes the transactional CONTINUUM primitives through dedicated `ioctl` dispatch codes. Legacy `cudaMalloc` and context-management paths remain untouched.
- **User-space runtime:** Implements Elastic Tensor operators, page-handle bookkeeping, and policy composition over the kernel interface.
- **Framework integration:** A thin integration layer that hooks the PyTorch/vLLM memory-management path, allowing existing kernels such as cuBLAS and FlashAttention to operate on ordinary virtual addresses without modification.

**Baselines.** We first evaluate the efficacy of our low-level engine by comparing it against CUDA VMM APIs. This baseline utilizes `cuMemMap` APIs to represent standard hardware-assisted memory management, serving as a canonical reference to quantify the control-plane overhead of vendor-provided drivers.

Subsequently, we evaluate CONTINUUM against several distinct system baselines:

- **PyTorch (Paszke et al., 2019):** Represents the traditional contiguous allocation strategy without paging support.
- **vLLM (Kwon et al., 2023):** The state-of-the-art serving system based on PagedAttention, representing the software-defined paging paradigm.
- **S-LoRA (Sheng et al., 2024):** A specialized system designed for heterogeneous workloads, which utilizes custom Triton kernels to manage non-contiguous memory.
- **vAttention (Prabhu et al., 2025):** A LLM inference framework which leverages standard VMM APIs to support inference workloads.

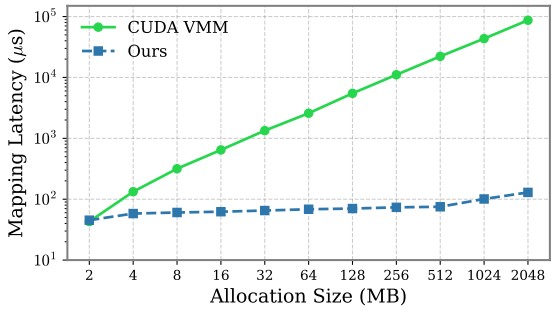

*Figure 6.* Mapping cost comparison between CONTINUUM and standard CUDA VMM.

## 4.2. Micro-benchmark

We begin by evaluating the latency of the CONTINUUM VMM backend when mapping physical memory to virtual addresses, using the 2 MB page granularity standard adopted in CUDA VMM API implementations. As shown in Figure 6, standard CUDA VMM exhibits linear-scaling behavior: mapping latency increases proportionally with the number of pages due to serialized driver interactions, quickly reaching prohibitive millisecond-level overheads. In contrast, CONTINUUM maintains near-constant, microsecond-level cost regardless of the allocation size. By decoupling PTE construction from the critical control path and leveraging batched DMA transactions, CONTINUUM achieves a speedup of up to **671.3×** compared to standard CUDA VMM APIs, reducing the mapping costs from the millisecond level to the microsecond level. This breakthrough performance enables the flexible, low-latency memory orchestration demanded by dynamic workloads.

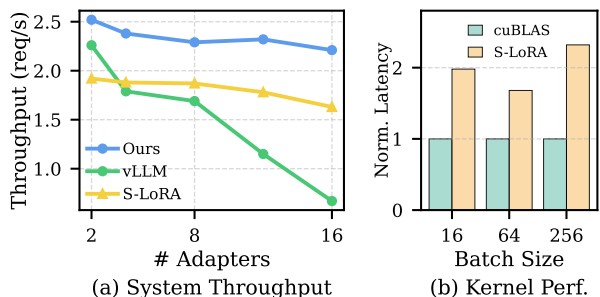

(a) System Throughput     (b) Kernel Perf.

*Figure 7.* Performance Comparison on Multi-LoRA Serving.

## 4.3. Multi-LoRA Inference

We evaluate CONTINUUM under heterogeneous Multi-LoRA serving workloads on Llama-2-7B (Touvron et al., 2023) [1] and compare against vLLM and S-LoRA. We synthesize the requests from ShareGPT (ShareGPT, 2023) and use a dummy LoRA adapter with rank 16. To stress-test elastic memory management, we run a simple simulation: each incoming request is assigned a random adapter ID, uniformly sampled from the active adapter set. vLLM starts with its default configuration of one adapter slot. We vary the number of active adapters from 1 to 16 to evaluate how systems behave under increasing oversubscription pressure.

As shown in Figure 7(a), CONTINUUM outperforms vLLM by up to 3.29×. Once vLLM's fixed adapter slots are exhausted, it incurs severe CPU–GPU swapping stalls; CONTINUUM eliminates this overhead by elastically mapping adapters to fragmented physical pages on demand, keeping GPU utilization high.

---

[1]Unless otherwise specified, all subsequent evaluations use Llama-2-7B.

Compared to S-LoRA, CONTINUUM achieves a 1.16–1.32× speedup. The gap mainly reflects backend choice: S-LoRA adapts LightLLM (ModelTC) and uses Triton, paying additional kernel-programming overhead, whereas CONTINUUM keeps elastic multi-adapter management but dispatches to cuBLAS (NVIDIA, 2024) for GEMM, as shown in Figure 7(b).

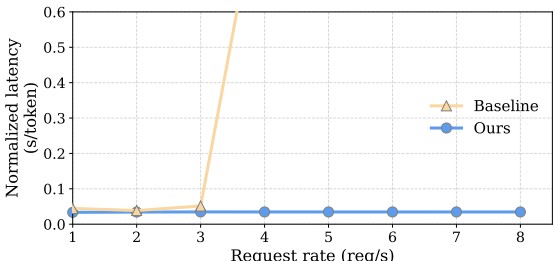

Figure 8. Performance comparison between the official copy-based implementation and CONTINUUM.

## 4.4. Adapting to Dynamic Model Architectures

A primary design goal of CONTINUUM is to deliver agile and efficient support for rapidly evolving model architectures. To test this on a workload outside the systems we co-developed, we benchmark CONTINUUM on Engram (Cheng et al., 2026), a recently proposed sparse retrieval-augmented architecture that exercises non-uniform memory access while still expecting a contiguous tensor view. Since the official open-source repository (DeepSeek-AI, 2026) serves primarily as a lightweight demonstration with mock computations, we adapt the workload configuration to align with the DeepSeekMoE-16B (DeepSeek-AI et al., 2024) architecture, so that the comparison reflects the memory pattern of a modern MoE model.

As shown in Figure 8, CONTINUUM achieves significantly higher throughput. The gap stems from a fundamental difference in memory management. Standard baseline frameworks, exemplified by PyTorch, are constrained by a *contiguous allocation requirement*. This forces *eager and monolithic allocation* of dynamic tensors (e.g., KV caches and embedding buffers), pre-provisioned for the maximum sequence length. The rigidity leads to severe memory overprovisioning and fragmentation, particularly as batch sizes scale, ultimately degrading throughput. In contrast, CONTINUUM employs *virtual contiguity*: it dynamically remaps disjoint physical blocks into a contiguous virtual range, so compute kernels operate directly on sparse data without explicit data movement, eliminating memory waste and maximizing throughput under heavy load.

*Table 2.* End-to-end latency speedup over vLLM at batch size 1 under varying branch factors ($K$) and step sizes ($L_{\text{step}}$).

| $K$ | $L_{\text{step}}$ (tokens between branching events) | | | | |
| --- | --- | --- | --- | --- | --- |
| | 1 | 4 | 16 | 64 | 256 |
| 2 | 0.97× | 1.42× | 1.32× | 1.29× | 1.23× |
| 3 | 0.99× | 2.04× | 1.87× | 1.76× | 1.50× |
| 4 | 1.03× | 2.85× | 2.50× | 2.31× | 1.87× |
| 8 | 1.03× | 7.65× | 6.23× | 5.34× | 3.64× |

## 4.5. Complex Reasoning Patterns

Advanced reasoning paradigms (e.g., Tree-of-Thoughts) introduce non-linear decoding structures characterized by frequent state branching and context inheritance. We evaluate this via tree-structured decoding parameterized by branch factor $K$ and step size $L_{\text{step}}$ (tokens generated between branches). Table 2 reports the batch-size-1 end-to-end latency speedup of CONTINUUM over vLLM.

At $L_{\text{step}}=1$, CONTINUUM performs on par with vLLM (0.97×–1.03×). Under this batch-size-1 regime, the highly constrained per-step computation inherits insufficient operational intensity to amortize the control-plane invocation overhead during token-by-token branching. Conversely, at moderate intervals (e.g., $L_{\text{step}}=4$), CONTINUUM achieves significant speedups, peaking at 7.65× for $K=8$. Under these regimes, software-paged baselines incur severe performance penalties from repeatedly reconstructing branch states, updating logical block tables, and managing physical memory fragmentation. In contrast, CONTINUUM resolves state inheritance entirely via zero-copy address-space aliasing at the MMU level. As $L_{\text{step}}$ expands to 256, the performance gains contract because branching events become sparse, allowing baselines to amortize their software-table management overheads across longer linear decoding chunks.

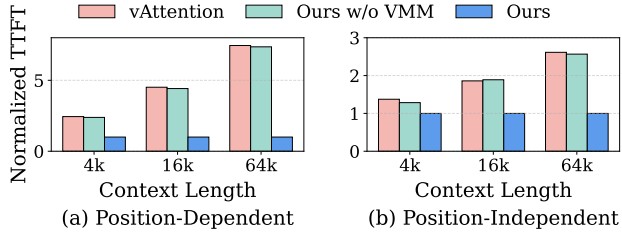

(a) Position-Dependent    (b) Position-Independent

*Figure 9.* Performance comparison on context caching.

## 4.6. Context Caching

We evaluate the efficacy of CONTINUUM in scenarios involving frequent reuse of substantial system prompts—a critical optimization for reducing Time-to-First-Token (TTFT). In this setting, the memory system should map massive context

states on-demand, accommodating both *Position-Dependent* and *Position-Independent* patterns (detailed in § 2). We vary the cached context length from 4K to 64K tokens. We benchmark CONTINUUM against **vAttention**, which utilizes standard `cuMemMap` APIs without driver-level optimizations. Additionally, we conduct an ablation study to isolate the gains attributable to our custom VMM backend and to quantify the raw overhead of the standard CUDA VMM control plane. While overlap scheduling can effectively mask mapping latency during the iterative decoding phase, it typically fails during instantaneous prefix restoration. Consequently, VMM mapping overhead sits squarely on the critical path, becoming the primary bottleneck for TTFT. As illustrated in Figure 9, standard VMM approaches (represented by vAttention and our ablation) incur prohibitive latency as context length increases—an overhead that paradoxically often exceeds the actual GPU compute time required to process the retrieved tokens. In contrast, CONTINUUM leverages its transactional VMM interface to minimize this cost.

## 5. Conclusion

We introduce CONTINUUM, a tensor virtualization middleware that bridges the gap between static framework assumptions and dynamic inference needs. The key insight is that a major obstacle to using hardware GPU VMM on the inference critical path lies not in the remapping mechanism itself, but in the latency and rigidity of its vendor-controlled control plane. By recasting this control plane as a transactional, batched remapping path inside the GPU driver, CONTINUUM turns hardware remapping into a low-latency runtime primitive. Built atop this primitive, the *Elastic Tensor* abstraction preserves a logical contiguous tensor view for unmodified, vendor-optimized kernels while enabling on-demand materialization and zero-copy aliasing. We see CONTINUUM as a candidate runtime layer for future serving stacks, complementary to higher-level scheduling and placement policies.

## Acknowledgements

We thank the anonymous reviewers for their thoughtful and constructive feedback, which substantially improved this paper. This work is supported by the National Key Research and Development Program of China (2024YFB4505701) and the National Natural Science Foundation of China (62232015, 62302479).

## Impact Statement

This paper presents work whose goal is to advance the field of machine learning. There are many potential societal consequences of our work, none of which we feel must be specifically highlighted here.

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

# A. Limitations

Currently, CONTINUUM focuses exclusively on optimizing the control-plane efficiency of device-resident GPU memory. We do not actively manage host memory (CPU RAM) for *automatic* oversubscription or paging (e.g., as found in CUDA UVM). Consequently, the system reports an Out-of-Memory (OOM) error when physical GPU capacity is saturated, mirroring the behavior of standard `cudaMalloc`.

**Compatibility with Capacity Expansion Strategies.** However, it is crucial to note that CONTINUUM is fully compatible with existing capacity extension techniques, both automatic and manual:

- **Explicit Offloading:** In production environments, applications frequently rely on *explicit* host-device data movement (e.g., `cudaMemcpy`) to manage capacity manually. CONTINUUM preserves this abstraction entirely. Since CONTINUUM-backed Elastic Tensors expose standard virtual addresses, users can seamlessly offload tensors to CPU memory and reload them on demand using standard runtime APIs without modification.

- **Orthogonality to Swapping:** Automatic swapping mechanisms (like vLLM's CPU blocks or OS-level paging) are orthogonal to our contribution. While CONTINUUM minimizes the latency of *mapping* pages, swapping extends the effective *capacity*. These mechanisms are complementary; in fact, CONTINUUM's fast-mapping primitives could theoretically accelerate the re-mapping phase of future swapping systems.

# B. Related VMM-Based Systems

Table 3 positions CONTINUUM against the most directly comparable prior VMM-based systems. The key distinction is that prior work uses the vendor VMM control plane as is and only *hides* its cost (e.g., via overlap scheduling), whereas CONTINUUM accelerates the control plane itself, which is what makes remapping feasible on the request critical path.

*Table 3.* Positioning of CONTINUUM relative to prior VMM-based systems. Prior work uses the vendor control plane as is and only hides the cost; CONTINUUM accelerates the control plane itself, which is what makes remapping feasible on the request critical path.

| System | Accelerates ctrl plane | Critical-path use | Restores contiguous view |
|---|---|---|---|
| GMLake (Guo et al., 2024) | ✗ | ✗ | training pools |
| vAttention (Prabhu et al., 2025) | ✗ | ✗ | KV cache only |
| **CONTINUUM (ours)** | ✓ | ✓ | general tensors |

# C. Metadata Overhead Analysis

### C.1. Theoretical Framework

A potential concern regarding our "Reserve-Upfront, Map-On-Demand" strategy is the memory footprint required for the page table structures. Since CONTINUUM reserves a complete virtual address (VA) space for maximum potential tensor shapes, one might assume this incurs substantial metadata overhead.

Formally, the metadata overhead $M_{cost}$ for a given virtual address range $V_{range}$ can be modeled as:

$$M_{cost} = \frac{V_{range}}{P_{size}} \times S_{entry} \tag{1}$$

where:

- $P_{size}$ is the hardware page size (2MB).

- $S_{entry}$ is the size of a single Page Table Entry (typically 16 bytes for 64-bit addressing).

### C.2. Quantitative Evaluation

To demonstrate the negligible nature of this overhead, we calculate the requirements for two representative scenarios relevant to LLM serving.

**Scenario 1: Full Context Reservation (64GB).** Consider a scenario where CONTINUUM reserves a massive 64GB virtual address space to accommodate the KV cache for a long-context request. Using the formula above:

$$M_{64GB} = \frac{64 \times 1024 \text{ MB}}{2 \text{ MB}} \times 16 \text{ Bytes} = 32,768 \times 16 \text{ Bytes} \approx \textbf{512 KB} \tag{2}$$

Allocating a mere 512 KB of contiguous memory to manage the mapping for 64 GB of virtual space represents a metadata overhead ratio of approximately $1 : 131,072$.

**Scenario 2: Runtime Mapping Cost (1GB).** Similarly, when the system dynamically populates the PTE buffer to map a 1GB segment of physical memory (e.g., a batch of new tokens), the size of the required PTE buffer is:

$$M_{1GB} = \frac{1024 \text{ MB}}{2 \text{ MB}} \times 16 \text{ Bytes} = 512 \times 16 \text{ Bytes} = \textbf{8 KB} \tag{3}$$

This means a single 8KB system page is sufficient to describe the mapping for 1GB of data.

When compared to the memory capacity of modern data center GPUs, this overhead is statistically insignificant. For an NVIDIA A100 (80GB capacity), the metadata required to map the *entire* device memory using CONTINUUM occupies less than **640 KB**. This represents less than **0.0008%** of the total device memory. Consequently, the trade-off of maintaining a complete virtual address view is fully justified by the performance gains in zero-copy management and kernel transparency.

## D. Implementation Details

### D.1. Non-Intrusive Driver Extension

To bridge the gap between user-space flexibility and kernel-level performance, our implementation spans both the OS kernel and the user runtime. A primary design constraint was to ensure strict backward compatibility with the existing NVIDIA CUDA and driver stack.

We achieved this via a *Non-Intrusive Extension Methodology* within the NVIDIA Open GPU Kernel Modules. Specifically, we introduced a new set of dedicated `ioctl` dispatch codes. The driver's main entry point was instrumented to recognize these specific CONTINUUM opcodes (e.g., for batch mapping or reserve operations) and route them to our isolated handler. Crucially, this design is strictly additive:

- All legacy `ioctl` calls falling outside our specific range are passed through to the original NVIDIA handler without modification.

- We do not alter the logic of existing memory allocation paths (e.g., `cudaMalloc`) or GPU context management.

This isolation ensures that CONTINUUM coexists peacefully with standard CUDA applications. A single process can mix legacy CUDA allocations with CONTINUUM managed Elastic Tensors without conflict, as they operate in orthogonal regions of the virtual address space.

### D.2. Interoperability with the CUDA Ecosystem

One more feature of CONTINUUM's design is the *Orthogonality of Control and Data Planes*. CONTINUUM exclusively manages the *Control Plane,* specifically, the manipulation of Page Tables to establish virtual-to-physical mappings. It does not interfere with the *Data Plane* (instruction execution and memory access).

Once a mapping is established by the CONTINUUM, the GPU Memory Management Unit (MMU) handles address translation transparently. The hardware makes no distinction between a page mapped via standard `cuMemMap` and one mapped via CONTINUUM. Consequently, the entire ecosystem of upper-layer CUDA libraries and APIs remains compatible:

- **Compute Kernels:** Standard launch APIs like `cudaLaunchKernel` operate correctly on CONTINUUM-backed memory. Unmodified kernels (e.g., FlashAttention, cuBLAS GEMM) can read/write these addresses directly.

- **Memory Operations:** Standard data movement primitives, such as `cudaMemcpy` (Host-to-Device / Device-to-Device) and `cudaMemset`, function seamlessly.

This transparency allows developers to integrate CONTINUUM into existing engines (like vLLM) with minimal code changes, effectively treating CONTINUUM as a drop-in replacement for the memory allocator.

### D.3. Synchronization semantics.

Our current implementation commits all mappings before launching any kernel that dereferences them, so kernels never observe transient or partially updated mappings. Similarly, `release`/`unmap` performs the required TLB invalidation before returning the underlying physical pages to the allocator pool, preventing stale translations from being reused by concurrently running streams.

## E. Portability

We summarize the prototype scope and what is required to port CONTINUUM to other hardware. The boundary is intentionally drawn so that the Elastic Tensor abstraction does not change across backends; only a localized backend driver layer does.

**Prototype scope.** The results in the main paper are obtained on a single NVIDIA A100 SXM GPU, CUDA 12.6, PyTorch 2.6, and Intel Xeon Gold 6430 host CPU. The default mapping granularity is 2 MB. At framework initialization we allocate the physical memory pool and pre-compute the corresponding PTE metadata; at request time we reserve the virtual tensor range and materialize it on demand.

The Elastic Tensor abstraction (`elastic`/`materialize`/`share`/`release`) and the transactional VMM design (eager reservation, offline PTE pre-calculation, batched commit) follow directly from the standard GPU MMU translation model and are not A100-specific. What *is* prototype-specific is the backend driver layer: ioctl dispatch, PTE encoding, and TLB invalidation semantics. Moving to a newer NVIDIA generation or a newer driver version requires localized backend adaptations rather than redesigning the abstraction.

**Generational portability on NVIDIA (Hopper H20).** To support this claim empirically, we have ported CONTINUUM to the Hopper architecture (NVIDIA H20). Table 4 compares mapping overhead between native CUDA VMM and CONTINUUM on H20 at the same 2 MB granularity used on A100. CONTINUUM preserves a consistent multi-order-of-magnitude speedup over native CUDA VMM, providing evidence of cross-generational portability.

*Table 4.* Mapping overhead on NVIDIA H20 (Hopper). CONTINUUM ports across NVIDIA generations without redesign and preserves its order-of-magnitude advantage over the native CUDA VMM control plane.

| Mapping size | CUDA VMM | CONTINUUM (ours) |
| --- | --- | --- |
| 32 MB | 1.07 ms | 0.061 ms |
| 128 MB | 3.78 ms | 0.085 ms |
| 512 MB | 14.56 ms | 0.118 ms |
| 1 GB | 31.14 ms | 0.127 ms |

## F. Security and Isolation Model

Given that CONTINUUM allows user-space runtimes to influence page table construction, ensuring security is paramount. We enforce a *Kernel-Mediated Trust Model* that preserves strict GPU security boundaries.

**1. Kernel-Resident Metadata.** All authoritative state regarding physical memory ownership is maintained exclusively in OS kernel space. The *Physical Page* and the *Staging PTE Buffers* are allocated in kernel-protected memory regions. User-space applications never access raw physical addresses (PAs) or modify the hardware page tables directly.

**2. Opaque Handle Abstraction.** The interface exposes only *Opaque Handles* to the user runtime. An opaque handle is a sanitized token representing a reference to a physical page frame. It does not leak information about the actual physical memory layout.

**3. Transactional Validation.** Every operation initiated by the user (e.g., `pmMemMap`) is treated as an untrusted request. The CONTINUUM driver extension performs mandatory validation before committing any changes to hardware:

- **Ownership Check:** The driver verifies that the provided opaque handles belong to the calling process's protection

domain.

- **Bounds Check:** The driver ensures that the target virtual address range is reserved and owned by the context.

- **Permission Enforcement:** Access permissions (Read/Write) are enforced by the kernel during PTE generation.

By acting as a mandatory gatekeeper, the CONTINUUM engine prevents privilege escalation, cross-tenant data leakage, and memory corruption, ensuring that the accelerated mapping path is as secure as the standard driver path.

## G. Interplay with Sub-Page Allocation

**Hardware Page.** Our hardware-assisted approach inherits the granularity constraints of the GPU MMU, which enforces a minimum page size of 4KB. Consequently, directly mapping tensors smaller than 4KB (e.g., small bias vectors) to independent physical pages would result in unavoidable internal fragmentation.

**Synergy with PyTorch's VMM Allocator.** To mitigate this, CONTINUUM is designed to adopt a cooperative model with existing ecosystem allocators. Notably, recent PyTorch versions have introduced `expandable_segments`, which also leverages CUDA VMM primitives (`cuMemMap`) to reduce external fragmentation by dynamically resizing memory pools. CONTINUUM functions as a perfect complement to this mechanism, forming a hierarchical memory solution:

- **Sub-Page Packing (PyTorch's Role):** For micro-tensors or static weights, standard allocators (like PyTorch's BFC) excel at sub-page packing. Since PyTorch's new allocator already sits atop VMM-backed segments, it efficiently handles the "dusty corners" of small object fragmentation.

- **Elastic Virtualization (CONTINUUM's Role):** CONTINUUM focuses on dynamic, large-scale structures (e.g., KV caches, Activation buffers). While PyTorch uses VMM primarily for pool resizing, CONTINUUM exposes the *programmable* capabilities of VMM (such as aliasing and lazy loading) to the user.

By offloading sub-page management to PyTorch's mature allocator while handling complex elasticity via CONTINUUM, we achieve a comprehensive solution that combines fine-grained efficiency with zero-copy agility.

## H. Future Work

CONTINUUM currently targets single-GPU inference, but its decoupled design opens several directions for broader deployment. First, we plan to extend CONTINUUM to distributed and disaggregated serving. The Elastic Tensor abstraction naturally generalizes to Unified Virtual Addressing (UVA) and RDMA-attached memory, enabling tensors to retain a logical contiguous view even when their physical backing spans devices or remote memory pools. Realizing this vision, however, requires additional support for cross-device page-table coordination, mapping consistency, and prefill/decode disaggregation, which we leave to future work.

Second, we plan to address capacity limitations through host-memory oversubscription. By transparently offloading cold tensor pages to CPU memory and rematerializing them on demand, CONTINUUM could evolve into a hierarchical memory-management layer that combines high-speed GPU remapping with elastic capacity across GPU and host memory.

Finally, we plan to explore cross-vendor portability. The core design of CONTINUUM is conceptually platform-agnostic, as it relies on hardware virtual-address translation rather than NVIDIA-specific programming semantics. A practical port to AMD/ROCm or emerging NPU backends, however, requires substantial engineering around vendor-specific driver interfaces, page-table encodings, remapping APIs, and TLB invalidation semantics. ROCm's open software stack may lower this barrier, but we conservatively leave comprehensive cross-vendor support to future work.

