# OpenReview forum: "CONTINUUM: Restoring the Contiguous Tensor Abstraction Efficiently for Dynamic AI Workloads via Hardware Virtualization"
_ICML.cc/2026/Conference — ICML 2026 spotlight_

### Official Review · Reviewer_CK7V · 2026-03-07

**Soundness:** 3
**Presentation:** 2
**Significance:** 2
**Originality:** 3
**Overall Recommendation:** 4
**Confidence:** 2

**Summary:**

This paper builds a new GPU memory virtualization layer to restore software-paged tensors to contiguous tensors. Compared to vAttention, it supports more dynamic scenarios like multi-LoRA and prefix sharing, and accelerates the management through a set of optimizations. Experimental results show that CONTINUUM outperforms baselines significantly.

**Compliance With Llm Reviewing Policy:**

Affirmed.

**Final Justification:**

This paper proposes a new GPU memory virtualization layer that reconstructs software-paged tensors into contiguous tensors, showing improvements over prior work such as vAttention.

The rebuttal addressed my concerns by clarifying the motivation and providing more details on the ToT configurations, which clarified the evaluation and strengthened my confidence in the results.

Overall, given the improved clarity and solid contributions, I have raised my score.

**Key Questions For Authors:**

1. Could you elaborate more on the limitations of vAttention? In particular, what is the main bottleneck of vAttention, and why does this bottleneck arise?
2. Why does CONTINUUM achieve such large speedups over vLLM on the ToT benchmark? More details about the benchmark setup would be helpful, such as the context length at each node. A latency breakdown would also help explain the source of the gains.
3. For Fig. 10, why does the paper only compare against vAttention, but not PagedAttention or TokenAttention? Also, how does vAttention handle prefix cache restoration in this setting?
4. Since the paper compares against S-LoRA, which is built on top of SGLang, why not also compare directly with SGLang? SGLang has better support for hiding CPU overhead.

**Limitations:**

Yes

**Strengths And Weaknesses:**

Strengths
1. The proposed solution shows significant performance improvements in the evaluated workloads.
2. The design can benefit multiple scenarios, including multi-LoRA serving and prefix sharing.

Weaknesses
1. The paper is not easy to follow。 I am still confused with the motivation of this work. First, for applications such as prefix sharing, serving systems still need fine-grained memory management, such as reference counting, even when virtual memory management (VMM) is used. In this case, it is unclear why VMM is needed to enforce contiguous allocation.
Second, from the kernel development perspective, existing abstractions such as the block-sparse format in FlashInfer already support several scenarios, including paging, sparse attention, and cascade attention. For sparse attention and cascade attention, VMM still requires indirect memory access. Therefore, using VMM may introduce more implementation complexity than existing approaches.
Third, the discussion of related work (e.g., vAttention) is not detailed enough. I still cannot clearly understand the main limitation of these systems or the root cause of the limitation.
2. The experimental results are surprisingly strong, but the paper does not provide enough analysis, such as breakdowns. For example, Fig. 9(b) shows that CONTINUUM achieves up to 2x speedup over vLLM, which is surprising. The paper attributes the improvement mainly to CPU reference counting overhead, which suggests that CPU overhead accounts for nearly half of the latency. This observation is inconsistent with my experience. CPU overhead is usually much smaller, especially when overlapped scheduling is enabled. In addition, the paper does not describe the ToT benchmark setup in sufficient detail.
3. The scope of the paper may be difficult for the ICML audience to fully appreciate. It may be a better fit for architecture or system venues such as ASPLOS.

---

> ### Author Rebuttal · Authors · 2026-03-31
>
> Thank you for the detailed feedback. We address the abstraction boundary, baseline rationale, and source of the gains below.
>
> **1. Motivation / abstraction boundary / vAttention limitation.**
> We agree that the original presentation did not make this boundary sufficiently clear. Our novelty is not “using VMM” itself; prior work such as vAttention already shows that GPU VMM is promising. Our novelty is to make VMM practical on the latency-sensitive serving critical path by modifying the GPU driver and redesigning the control plane as a transactional, batched remapping path.
>
> We also agree that VMM does not remove logical bookkeeping such as reference counting, radix lookup, or cache identity management. These mechanisms still decide what to share. CONTINUUM plays a different role: once that decision is made, it exposes the shared/private composition as a single contiguous tensor view via remapping and aliasing, so existing kernels can operate on ordinary virtual addresses. Thus, VMM is not a replacement but decouples reference counting from kernel-visible layout.
> This also clarifies the limitation of vAttention. Its bottleneck is not missing logical sharing machinery, but continued reliance on the native vendor VMM control plane. As a result, it can mainly hide or amortize remapping overhead, but cannot remove the root cause: the high software overhead of fine-grained, synchronous, driver-mediated VMM remapping on the serving critical path. CONTINUUM targets this bottleneck directly.
>
> **2. Relation to FlashInfer / cascade attention.**
> We view these approaches as complementary rather than competing. Specialized paged/sparse kernels are highly effective when one is willing to redesign kernels for a specific operator. CONTINUUM addresses a different goal: restoring the standard contiguous tensor abstraction across broader dynamic memory patterns, including cases beyond attention, without rewriting kernels. We will revise the paper to make this scope distinction clearer.
>
> **3. Why compare against S-LoRA rather than SGLang?**
> This question goes to the core motivation of our abstraction design. We compare against S-LoRA (which is implemented atop LightLLM rather than SGLang) because it highlights a fundamental issue with software-paged kernels. S-LoRA requires non-contiguous multi-adapter execution through specialized page-aware kernels. Integrating the same capability directly into a framework like SGLang, which already tightly couples KV-cache paging, requires non-trivial engineering to co-manage both LoRA and KV-cache states at fine granularity (even though both projects were developed by the same open-source team). This illustrates the limitation of the software-paged paradigm.
>
> In contrast, CONTINUUM uses hardware-assisted virtual stitching to expose a contiguous memory view to unmodified dense kernels. This lowers the engineering barrier and allows new algorithms, such as multi-adapter serving, to be deployed without writing specialized page-aware kernels. We will make this deployment-agility argument clearer in the revision.
>
> **4. Why does Figure 10 compare to vAttention rather than PagedAttention?**
> Figure 10 is designed to isolate the control-plane cost of VMM-based context restoration, so we selected vAttention as the most direct VMM baseline. Comparing against PagedAttention or TokenAttention, which rely on software-managed block tables and specialized paged kernels, would conflate VMM mapping with software paging.
>
> We also emphasize that CONTINUUM is orthogonal to and compatible with general caching policies such as RadixAttention. In a modern serving stack, radix-style systems define the policy (which prefix states to reuse), while CONTINUUM provides the mechanism (how those reused states are materialized into contiguous tensor views without copying). We will make this policy/mechanism boundary explicit in the revision.
>
> **5. Why ToT speedup is so large?**
> The large speedup stems from eliminating the CPU overhead exposed by complex, dynamic branching.
> As noted by recent optimizations in inference engines (e.g., SGLang v0.4), CPU-side operations like memory allocation, KV-table updates, and prefix matching can consume up to 50% of inference time if unoptimized. In ordinary linear decoding, branch creation rarely occurs. Frameworks can aggressively overlap CPU/GPU execution, meaning VMM remapping is not on the critical path.
> However, in branch-structured decoding, the branching pattern is highly dynamic and data-dependent. The system must repeatedly branch and inherit states based on the specific output of previous steps. In software-paged systems (like vLLM), it forces a CPU-bound bottleneck: the engine must perform deep copies of tensor metadata and continuously update user-space block tables, which cannot be easily overlapped.
> CONTINUUM bypasses this bottleneck by reducing branch creation and state inheritance to fast, zero-copy virtual aliasing via hardware MMU metadata. Will clarify.

---

> > ### Author Rebuttal · Reviewer_CK7V · 2026-04-03
> >
> > Thank you for your response. Could you elaborate more on the configurations of ToT? I understand that for beam search, there will be frequent branching, but for ToT, the branching frequency will be much lower. Could you provide some statistical numbers on ToT, like context length of each node, how many branches, etc?

---

> > > ### Author Response · Authors · 2026-04-08
> > >
> > > We thank the reviewer for the valuable follow-up question. We agree that our original explanation of the tree-structured reasoning configuration was not sufficiently clear. We also want to clarify that **ToT/beam search is only an evaluation workload in our paper, not our primary contribution**; our goal is to evaluate how different branching patterns stress the underlying memory-management/runtime mechanism.
> > >
> > > The result currently reported in the paper (Figure 9(b), described around Line 434) corresponds to a **beam-search-style setting implemented at the decode level in vLLM**, a **branch size 2 with \(L=1\)** setup. We will revise the paper to make this scope explicit.
> > >
> > > In this sense, it is conceptually similar to ToT-style reasoning because it exercises repeated branching and state inheritance, but it is not intended as a verbatim implementation of any particular sequence-level ToT algorithm, especially since there is no single dominant implementation standard.
> > >
> > > We parameterize the workload by a **step size** \(L\): each active node generates \(L\) tokens before the next branching/pruning event. Thus, \(L\) directly determines the branching frequency.
> > > Under this parameterization, \(L=1\) reduces to the beam-search-style setting already used in the paper; more generally, as suggested by the reviewer, smaller \(L\) makes the workload more beam-search-like, while larger \(L\) makes it progressively closer to linear decoding with less frequent, more coarse-grained branching.
> > >
> > > We have now run additional experiments for \(L \in \{1, 4, 16, 64, 256\}\), and all newly added experiments below use **batch size 1** to isolate the effect of branching granularity. At larger batch sizes, we expect the behavior to follow the same trend already observed in **Figure 9**, where the performance gap widens as the system must maintain and recover more live branches/states simultaneously.
> > >
> > > As a preliminary demonstration, we report the following end-to-end latency results:
> > >
> > > | Branches | Step size *L* | vLLM (s) | Ours (s) | SpeedUp |
> > > | :--- | :--- | :--- | :--- | :--- |
> > > | 2 | 1 | 8.260 | 8.545 | 0.97x |
> > > | 2 | 4 | 10.513 | 7.408 | 1.42x |
> > > | 2 | 16 | 9.353 | 7.100 | 1.32x |
> > > | 2 | 64 | 8.991 | 6.949 | 1.29x |
> > > | 2 | 256 | 8.412 | 6.859 | 1.23x |
> > > | 3 | 1 | 9.033 | 9.084 | 0.99x |
> > > | 3 | 4 | 15.708 | 7.681 | 2.04x |
> > > | 3 | 16 | 13.638 | 7.300 | 1.87x |
> > > | 3 | 64 | 12.568 | 7.123 | 1.76x |
> > > | 3 | 256 | 10.593 | 7.068 | 1.50x |
> > > | 4 | 1 | 10.283 | 9.998 | 1.03x |
> > > | 4 | 4 | 22.861 | 8.024 | 2.85x |
> > > | 4 | 16 | 18.877 | 7.565 | 2.50x |
> > > | 4 | 64 | 17.003 | 7.354 | 2.31x |
> > > | 4 | 256 | 13.721 | 7.323 | 1.87x |
> > > | 8 | 1 | 15.664 | 15.174 | 1.03x |
> > > | 8 | 4 | 73.995 | 9.671 | 7.65x |
> > > | 8 | 16 | 56.861 | 9.122 | 6.23x |
> > > | 8 | 64 | 48.344 | 9.062 | 5.34x |
> > > | 8 | 256 | 32.325 | 8.882 | 3.64x |
> > >
> > > These results show two clear trends. First, for \(L=1\) (the beam-search-style decode-level setting already used in the paper), CONTINUUM is roughly on par with vLLM, consistent with the current results. Second, for larger \(L\), the gap grows because software-managed systems must repeatedly reconstruct branch state and restore context at each branching boundary, while CONTINUUM reduces this to zero-copy aliasing through MMU metadata. As \(L\) becomes very large, this fixed software overhead is increasingly amortized over longer generation chunks, so the relative speedup naturally decreases.
> > >
> > > We will revise the manuscript to make these configurations explicit, including the branch factor, step size \(L\), the node-context-length interpretation, and the distinction between the original beam-search-style decode-level setting and the larger-step ToT-style settings above.
> > >
> > > For reproducibility, we have released an anonymized repository with the current implementation and will open-source it upon acceptance: https://anonymous.4open.science/r/ICML-470-0481
> > >
> > > We thank the reviewer again for this valuable follow-up. We hope the additional clarification and results above address the concern, and we will make these configuration details explicit in the revision.

---

### Official Review · Reviewer_6acq · 2026-03-11

**Soundness:** 4
**Presentation:** 2
**Significance:** 4
**Originality:** 4
**Overall Recommendation:** 5
**Confidence:** 4

**Summary:**

This paper argues that emerging LLM inference workloads (e.g., long-context serving, prefix caching, multi-LoRA composition, and tree/branching reasoning) require “extreme memory agility” while still depending on the contiguous tensor abstraction to reuse highly optimized vendor kernels. The authors identify a key bottleneck that prevents simply using GPU hardware virtual memory to reconcile these needs: existing CUDA VMM mapping operations are too slow (millisecond-scale and scaling with mapping size), making fine-grained remapping infeasible on the inference critical path. To address this, the paper introduces CONTINUUM, a middleware/hardware-virtualization approach that (i) fundamentally accelerates GPU VMM mapping via driver-level transactional/batched primitives (by moving page-table metadata preparation off the critical path and committing mappings through batched DMA-style updates), and (ii) exposes an Elastic Tensor abstraction to frameworks so tensors remain virtually contiguous while being backed by fragmented physical pages, enabling low-overhead growth, sharing, and zero-copy aliasing. The evaluation integrates CONTINUUM with vLLM and shows large reductions in mapping latency as well as throughput/latency benefits on representative dynamic workloads including multi-LoRA inference, tree decoding, and context caching.

**Compliance With Llm Reviewing Policy:**

Affirmed.

**Key Questions For Authors:**

1. Implementation details and integration boundary.
Could the authors clarify the concrete implementation path of CONTINUUM end-to-end: which components are implemented in the kernel module vs. user-space runtime vs. framework integration (e.g., PyTorch/vLLM), and what exact interfaces are exposed at each boundary (including any required synchronization semantics with streams)?
How this affects my evaluation: If the system can be integrated with minimal kernel/driver surface area and clearly defined semantics, it strengthens my confidence in practical adoptability. If it requires substantial invasive driver changes or relies on assumptions that are hard to maintain across GPU generations, it would slightly reduce my assessment of practical impact (though not necessarily technical soundness).
2. Open-sourcing plan and reproducibility.
Do the authors plan to open-source (a) the user-space runtime and Elastic Tensor integration, and (b) the kernel-module changes (or a reproducible patch set) needed to expose the transactional VMM primitives? If not, can the authors provide an alternative reproducibility strategy (e.g., detailed artifacts, scripts, and a minimal kernel patch) that enables independent validation?
How this affects my evaluation: A credible open-source/reproducibility plan would strengthen my confidence in the work’s long-term impact and community uptake. If the critical components cannot be shared, it would not change my assessment of correctness, but it would reduce expected practical influence.

**Limitations:**

yes

**Strengths And Weaknesses:**

Soundness
Strengths: The submission is technically well grounded. The central bottleneck is clearly identified and validated with targeted microbenchmarks, and the end-to-end experiments are aligned with the claimed benefits (elastic growth, reduced software bookkeeping, and MMU-based zero-copy aliasing) in workloads where those benefits should matter. The system design is coherent (driver primitives + runtime + framework integration), and the reported results are consistent with the paper’s stated mechanism.
Weaknesses (minor / low-stakes): The empirical section would be slightly easier to interpret with more explicit reporting of a few operational details (e.g., mapping granularity defaults, reserve/materialize policies, and any warm-up behavior). These are not essential to believe the results, but they would help readers reproduce the exact conditions and understand sensitivity.

Presentation
Strengths: The paper is well structured and easy to follow. The “motivation → bottleneck → design → evaluation” progression is clear, and the RQ-based evaluation framing helps. Figures that illustrate the design space and the scaling behavior of legacy VMM mapping are particularly effective.
Weaknesses (cosmetic but fixable): The reference formatting is inconsistent and should be standardized. In addition, a short “limitations/portability” subsection would improve readability by explicitly summarizing what requires kernel-module support versus what could be adopted purely in user space or with future vendor support—this is mainly a documentation clarity issue rather than a technical gap.

Significance
Strengths: The problem is highly relevant to modern ML systems practice. Restoring contiguous tensor semantics while enabling memory agility and aliasing directly addresses a real pain point in LLM serving: preserving compatibility with highly optimized kernels while avoiding engineering-heavy paged-aware implementations. The techniques could plausibly influence how future serving systems and runtimes manage dynamic state and shared prefixes.
Weaknesses (minor framing): The paper’s impact is primarily in systems/runtime engineering for inference rather than general ML algorithms. This is appropriate for the contribution; it would just benefit from one or two sentences in the conclusion that more explicitly articulate the likely adoption path (e.g., as a runtime layer in serving stacks, or as a candidate direction for vendor-supported primitives).

Originality
Strengths: While hardware VMM exists in principle, the paper’s originality lies in making this route practically viable for dynamic inference by (i) removing the main obstacle (slow mapping control plane) via transactional/batched primitives and (ii) packaging it into a framework-facing Elastic Tensor abstraction that preserves contiguous semantics while supporting elastic growth and zero-copy aliasing. This is a meaningful, non-trivial combination with clear practical payoff.
Weaknesses (minor positioning): Because the underlying mechanism (virtual memory) is familiar, some readers may initially underestimate novelty. A small presentation tweak—more explicitly stating “what becomes feasible only after transactional mapping” versus “what was already possible in principle”—would make the novelty easier to appreciate without changing any technical content.

---

> ### Author Rebuttal · Authors · 2026-03-31
>
> Thank you for the very positive assessment and for the concrete suggestions on implementation boundary, reproducibility, portability, and presentation. We agree these clarifications would make the paper stronger.
>
> **1. Implementation details / interfaces / synchronization semantics.**
> Thank you for asking for a clearer end-to-end implementation path. We agree that the implementation boundary should be made more concrete and will clarify this in the revision. CONTINUUM is split into three layers:
>
> - **Kernel module:** reserve/create/map/unmap-style transactional VMM primitives, validation, and the page-table commit path.
> - **User-space runtime:** Elastic Tensor operations (elastic, materialize, share, release), page-handle bookkeeping, and policy-level composition.
> - **Framework integration:** the PyTorch/vLLM memory-management layer that exposes ordinary virtual addresses to existing kernels/libraries.
>
> We will also add a short paragraph on synchronization semantics: mappings are committed before launching kernels that dereference them, and release/unmap performs the necessary invalidation before physical reuse.
>
> **2. Driver surface / adoptability.**
> Thank you for highlighting adoptability. We agree that this depends strongly on how invasive the driver changes are.
>
> Our implementation is intentionally additive: we introduce dedicated transactional VMM `ioctl`s without modifying legacy allocation paths or context-management behavior. We will make this “additive rather than replacing” design choice much more explicit.
>
> **Open-source / reproducibility.**
> Thank you for asking about reproducibility. We agree that making this plan explicit will strengthen the paper and will add it to the revision.
>
> After de-anonymization, we plan to release the full user-space runtime, the framework integration, and the kernel-module patch set against the NVIDIA open kernel modules.
>
>
> **3. Portability / limitations / operational details.**
> Thank you for suggesting a clearer discussion of portability and limitations. We agree that adding a short dedicated subsection would improve readability.
>
> Our current prototype is evaluated on an A100/CUDA/PyTorch/vLLM stack, but the underlying abstraction remains independent of the specific backend engineering required for the system. In the final version, we will make this boundary explicit and clearly distinguish what is prototype-specific from what follows directly from the abstraction itself. Concretely, we use a default mapping granularity of 2 MB. We reserve the virtual tensor range at initialization and materialize it on demand, while the physical memory pool and the corresponding PTE metadata are created during framework initialization.
>
> In addition, we have recently ported CONTINUUM to the Hopper architecture (NVIDIA H20). The microbenchmark below compares memory-mapping overhead between native CUDA VMM and CONTINUUM on H20:
>
> | Size | CUDA VMM | Ours |
> |---|---:|---:|
> | 32 MB | 1.07 ms | 0.061 ms |
> | 128 MB | 3.78 ms | 0.085 ms |
> | 512 MB | 14.56 ms | 0.118 ms |
> | 1 GB | 31.14 ms | 0.127 ms |
>
> More broadly, while the core design is conceptually platform-agnostic, a practical port to alternative vendor stacks (e.g., AMD’s ROCm) would entail substantial engineering effort to accommodate vendor-specific driver interfaces, PTE formats, and mapping semantics. Although the open-source nature of ROCm facilitates such integration, we consider comprehensive cross-vendor support to be a promising direction for future work.
>
> **4. Making the novelty clearer.**
> Thank you for this suggestion. We fully agree that because VMM itself is a familiar mechanism, the paper should more explicitly highlight what becomes feasible only after fast transactional remapping.
>
> The contribution is not “using VMM in principle”, but making remapping sufficiently fast to act as a runtime primitive, and then exposing it as an Elastic Tensor abstraction that enables contiguous-view execution together with zero-copy aliasing, prefix reuse, and dynamic tensor growth. We will make this distinction more explicit in both the related-work discussion and the conclusion.

---

> > ### Author Rebuttal · Reviewer_6acq · 2026-04-02
> >
> > My concerns are fully resolved. The rebuttal clearly explains the implementation split across kernel module, user-space runtime, and framework integration, and it also clarifies synchronization semantics. The authors further addressed adoptability by explaining that the driver changes are additive rather than replacing legacy paths.
> >
> > I also appreciate the explicit reproducibility plan, as well as the added operational details such as mapping granularity, reserve/materialize behavior, and initialization policy. The additional Hopper/H20 results are also helpful.
> >
> > Overall, these responses adequately address the main clarification points I raised in my original review, which were mostly about implementation boundary, reproducibility, portability/limitations, and presentation of novelty.

---

> > > ### Author Response · Authors · 2026-04-08
> > >
> > > We thank the reviewer again for their valuable follow-up. We have released an anonymized repository containing our current implementation, which will be fully open-sourced upon acceptance: https://anonymous.4open.science/r/ICML-470-0481.  As the codebase is still undergoing cleanup, some legacy project names still appear in certain parts of the source tree. We are actively refining this to enhance readability. For your convenience, the key engineering changes within the GPU driver are primarily located in `driver-patch/kernel-open/nvidia-uvm/uvm_map_external.c`.  We would greatly appreciate any further guidance.

---

### Official Review · Reviewer_J9Cc · 2026-03-13

**Soundness:** 3
**Presentation:** 2
**Significance:** 3
**Originality:** 3
**Overall Recommendation:** 5
**Confidence:** 3

**Summary:**

This paper proposes and implements a GPU memory management technique tailored for AI workloads. In recent LLM inference systems, fine-grained and dynamic memory management has become essential in order to efficiently utilize limited GPU memory. Currently, fragmentation is avoided through user-level software paging, but this approach sacrifices software transparency. The existing CUDA Virtual Memory Management (VMM) implementation is optimized for coarse-grained memory allocation and incurs large mapping overheads. To address this issue, the authors propose a new approach called CONTINUUM, which works in conjunction with a kernel driver. In the evaluation, the performance of the proposed system is compared with existing CUDA VMM and the software implementation used in vLLM.

**Compliance With Llm Reviewing Policy:**

Affirmed.

**Final Justification:**

After reading the rebuttal, I have increased my score to Accept. My primary concerns have been well addressed through the authors’ careful responses. This paper tackles an important system-level problem. While the topic may be more closely aligned with computer systems conferences such as ASPLOS, the proposed idea and its implementation are practical and valuable. I believe this type of work should be encouraged within the machine learning community. Therefore, I support acceptance.

**Key Questions For Authors:**

In HPC workloads on CPUs, programmers often allocate a large memory region in advance and reuse that region at the user level in order to avoid the overhead of dynamic memory allocation. Could ML workloads on GPUs also rely on similar user-level control instead? In particular, for developers implementing ML kernels who want to push performance optimization to the limit, would manual memory management be preferable to driver-level automatic memory management, since it may allow easier performance tuning?

Even if we limit the discussion to NVIDIA GPUs, what kinds of modifications or considerations would be required to make the CONTINUUM implementation usable on newer generations of GPUs? Is it feasible for the proposed technique to be incorporated into the NVIDIA GPU driver?

**Limitations:**

Yes

**Strengths And Weaknesses:**

Strengths:

This work identifies problems in current GPU memory management techniques and proposes a new method suitable for dynamic memory management in ML workloads. The approach is implemented in practice and its efficiency is demonstrated, making this a practical study. The engineering effort, including extensions to the GPU driver, is substantial, and the practicality of the developed software technology deserves high recognition.

Weaknesses:

The paper does not analyze to what extent dynamic memory management introduces performance overhead in today’s typical ML workloads. Figure 2 evaluates mapping size and latency; however, the authors should first quantitatively evaluate how significant this mapping cost is as a performance bottleneck in real ML workloads.

The proposed method improves latency by removing PTE allocation from the critical path, but this requires additional precomputation. It is not clearly explained how the overhead of this precomputation is hidden. While the proposed virtual memory management scheme is practical, the idea itself is not particularly surprising and appears to rely largely on engineering effort.

CONTINUUM is implemented targeting the NVIDIA A100 GPU, but it is unclear whether the ideas can be applied to newer-generation GPUs such as the H100 or to GPUs from other vendors. The paper should clearly specify the functional requirements that the GPU and the controlling CPU must provide.

Typo:

l.186, Worload -> Workload

---

> ### Author Rebuttal · Authors · 2026-03-31
>
> Thank you for the thoughtful review. Below we clarify when mapping cost matters, how precomputation is accounted for, and the portability boundary.
>
> **1. How significant is mapping cost in real ML workloads?**
> We agree that this point should be stated more precisely. Our claim is not that mapping overhead dominates every ML workload, but that it becomes critical when remapping cannot be overlapped and thus lies on the critical path. This is reflected in our results: in linear decoding, CONTINUUM largely matches vLLM (with only a modest long-context gain), whereas in tree decoding the gain rises substantially because branch inheritance lies on the critical path. Likewise, in context caching, the issue is instantaneous prefix restoration, where overlap scheduling is ineffective and standard VMM latency can exceed the actual GPU compute for the retrieved tokens. We will make this “critical-path vs. overlappable” distinction much clearer in Section 4.
>
> **2. How is the precomputation overhead hidden?**
> Thank you for raising this point. We agree this needs a clearer explanation.
>
> The key point is that PTE construction is not performed during latency-sensitive remapping. It is performed once when physical pages are created/admitted into the pool (`MemCreate`), and the resulting entries are stored in a persistent host-side PTE buffer. As a result, critical-path `MemMap` is reduced to committing precomputed metadata, rather than constructing per-page mappings at request time. In other words, the cost is not eliminated; it is moved off the latency-sensitive remapping path and amortized at page admission / pool-preparation time. We will clarify this execution timeline and add warm-up/pool-preparation details in the revision.
>
> **3. Why not rely on manual/user-level memory management instead?**
> We agree this distinction between manual pooling and a systems-layer solution should be stated more clearly.
>
> For narrow and highly specialized workloads, expert developers can often obtain good performance from hand-tuned allocators and custom kernels. Our goal is different: to show that there is a missing systems layer that preserves high performance while greatly reducing kernel/programming burden under rapidly evolving dynamic workloads. CONTINUUM does not remove workload-aware policies from user space; rather, it restores the option of using standard contiguous kernels after those policies decide placement and sharing. This is the key distinction between manual pooling and restoring contiguity at the virtualization layer.
>
> **4. CPU / portability / Could this be incorporated into the NVIDIA driver / newer generations?**
> We agree the portability, deployment, and upstream-integration boundaries of our system should be stated more clearly.
>
> **Generational portability on NVIDIA.** Fundamentally, CONTINUUM relies on standard hardware virtual-address translation (i.e., GPU MMU capabilities), not on A100-specific semantics. This excludes legacy architectures without virtual memory support, but supports portability across modern NVIDIA generations. Moving to a newer generation mainly requires localized backend adaptations, e.g., driver interfaces, PTE encodings, and invalidation semantics without redesigning the core Elastic Tensor abstraction. Our current implementation is intentionally additive. We introduce dedicated `ioctl`s and handlers for CONTINUUM’s new primitives without modifying legacy `cudaMalloc` paths or GPU context management. This makes the prototype a practical extension of the existing stack rather than a monolithic fork. We will revise the manuscript to make this “same abstraction, different backend” boundary explicit.
>
> **Empirical validation on Hopper.** The current paper evaluates one concrete instantiation on NVIDIA A100 (CUDA 12.6, PyTorch 2.6) with an Intel Xeon Gold 6430 CPU, and we will state this scope more explicitly in the revision. At the same time, to further validate portability, we have recently ported CONTINUUM to Hopper (NVIDIA H20). The microbenchmark below compares mapping overhead between native CUDA VMM and CONTINUUM on H20:
>
> | Size | CUDA VMM | Ours |
> |:---:|:---:|:---:|
> | 32M  | 1.07 ms | 0.061 ms |
> | 128M | 3.78 ms | 0.085 ms |
> | 512M | 14.56 ms | 0.118 ms |
> | 1G   | 31.14 ms | 0.127 ms |
>
> These results show that CONTINUUM preserves large/consistent speedup over native CUDA VMM on Hopper, providing evidence of cross-generational portability.
>
> **Cross-vendor portability (AMD/ROCm).** More broadly, the core design is conceptually platform-agnostic. However, a practical port to another vendor stack still requires substantial engineering effort to adapt to vendor-specific driver interfaces, PTE formats, and mapping semantics. While the openness of ROCm lowers the barrier to entry, we believe it is more accurate to frame cross-vendor support as future work.
>
> Finally, we thank the reviewer for pointing out the typo, which we will correct in the revision.

---

> > ### Author Rebuttal · Reviewer_J9Cc · 2026-04-04
> >
> > Thank you for the clear and thorough responses to my questions. My concerns have been resolved, and I will consider revising my score.

---

> > > ### Author Response · Authors · 2026-04-08
> > >
> > > We thank the reviewer again for the valuable guidance. We have released an anonymized repository containing our current implementation, which will be fully open-sourced upon acceptance: https://anonymous.4open.science/r/ICML-470-0481. As the codebase is still undergoing cleanup, some legacy project names still appear in certain parts of the source tree. We are actively refining this to enhance readability. We would be very grateful if the reviewer continues to view the work favorably in the final discussion.

---

### Official Review · Reviewer_jjox · 2026-03-18

**Soundness:** 3
**Presentation:** 3
**Significance:** 4
**Originality:** 2
**Overall Recommendation:** 5
**Confidence:** 3

**Summary:**

This work identifies a key tension between deep learning frameworks (eg PyTorch) that rely on contiguous tensor addressing (and can rely on vendor libraries) but lead to memory fragmentation versus inference serving frameworks, such as vLLM, that leverage paged tensors, but require custom page-aware kernels.

To address this, the authors introduce "transparent tensor virtualization", a middleware between the deep learning framework and hardware-level memory primitives. The key contribution is a high-performance backend implementation of the CUDA Virtual Memory Management (VMM) APIs that leverage the GPU's native MMUs.

**Compliance With Llm Reviewing Policy:**

Affirmed.

**Key Questions For Authors:**

I liked this paper a lot - and I see a lot of value in the overall direction. Leveraging the GPU's native MMU and enhancing it for Paged Kernels makes a lot of sense.

I have three sets of questions/concerns.

(1) Novelty.
It would help to add some form of related work table to clearly articulate the novelty of this work over prior works (eg vAttention) that have also proposed using VMMs. I see some discussions in the eval section but it'll be valuable to make this more explicit.

(2) Extensibility
The implementation serves as a valuable proof-of-concept, but I am curious how specific the changes were to the PyTorch version, CUDA version and GPU generation? I am also curious if you see this idea extending beyond NVIDIA GPUs to say AMD GPUs?
I also see SGLang mentioned in the motivation as vLLM (which makes sense for a general motivation) - though I am not entirely sure if the authors have thought through if/how SGLang's Radix Attention mechanism would play with this?

(3) Evaluations
There is growing interest in studying vLLM for agentic deployments, prefill-decode disaggregated setups etc. Would CONTINUUM work seemlessly there?

**Limitations:**

yes

**Strengths And Weaknesses:**

Strengths:
+ removes the need for paged kernels, enhancing productivity
+ valuable engineering to re-implement the VMM driver.

Weaknesses:
- The idea of leveraging VMMs has been proposed previously by vAttention
- Unclear if/how the implemenation would changes as PyTorch, CUDA and GPU HW evolves.

---

> ### Author Rebuttal · Authors · 2026-03-31
>
> Thank you for the positive assessment and for highlighting the importance of clarifying novelty, extensibility, and applicability to broader serving settings.
>
> **1. Novelty relative to vAttention / GMLake / prior VMM-based work.**
> We agree that the paper should make this distinction much more explicit. Our claim is not that “using VMM” is itself novel. We view **vAttention** [ASPLOS'25] as the serving-side line of prior VMM-based systems and **GMLake** [ASPLOS'24] as the training-side line. Both are important evidence that GPU VMM is a promising direction. However, they still rely on the vendor VMM control plane as-is, and therefore reduce overhead mainly by hiding or amortizing it (e.g., overlap scheduling, premapping, or coarser-grained management), rather than reducing the root bottleneck itself.  **Our novelty is in making GPU VMM practical on the latency-sensitive serving critical path by modifying the GPU driver and changing the control plane to a transactional, batched remapping path.**
>
> We agree that this should be positioned more clearly. In the revision, we will add a related-work table that explicitly contrasts:
> (a) whether a method merely uses vendor VMM as-is vs. accelerates the VMM control plane itself,
> (b) whether remapping is feasible on the request critical path, and
> (c) whether the system restores contiguous tensor semantics beyond a single attention-specific or workload-specific use case.
>
> **2. Extensibility / PyTorch, CUDA, GPU generation / AMD / SGLang.**
> Thank you for raising these portability questions. We agree they are important for judging long-term impact and practical adoption.
>
> The current paper evaluates one concrete instantiation on NVIDIA A100 (CUDA 12.6, PyTorch 2.6).
> We will clarify the implementation boundary more concretely. In kernel space, we extend the NVIDIA open kernel modules with an additive transactional VMM surface. In the user space, the CONTINUUM runtime manages Elastic Tensor operations (elastic, materialize, share, release) and invokes the kernel interface. At the framework level, we integrate this runtime into vLLM/PyTorch so that existing kernels (e.g., cuBLAS / FlashAttention) operate on ordinary virtual addresses without modification. Porting to a new generation does not require a redesign of the Elastic Tensor abstraction.
>
> More broadly, CONTINUUM relies on hardware virtual-address translation rather than A100-specific behavior. As a result, very old GPUs without virtual memory support cannot support it, but the abstraction is portable across modern NVIDIA generations. We will make this “same abstraction, different backend” boundary explicit.
> To further support this claim, we have recently ported CONTINUUM to Hopper (NVIDIA H20). The microbenchmark below compares memory-mapping overhead against CUDA VMM:
>
> | Size | CUDA VMM | Ours  |
> |---:|---:|---:|
> | 32 MB | 1.07 ms | 0.061 ms |
> | 128 MB | 3.78 ms | 0.085 ms |
> | 512 MB | 14.56 ms | 0.118 ms |
> | 1 GB | 31.14 ms | 0.127 ms |
>
> These results show that we maintain a consistent speedup over the native CUDA VMM on the Hopper architecture.
> For AMD, although the design is conceptually platform-agnostic and ROCm lowers the barrier to entry, a practical port still requires substantial engineering around vendor-specific backend driver adaptation (e.g., driver interfaces, PTE formats, invalidation semantics). We therefore frame cross-vendor support as future work.
>
> Regarding SGLang / Radix Attention, we agree this deserves clearer discussion. At the abstraction level, CONTINUUM is compatible with radix-style prefix sharing: a matched prefix can be re-exposed as aliased physical pages inside a newly reserved contiguous virtual range, followed by materialization of the private suffix. In that sense, CONTINUUM is complementary to radix-based lookup structures: the lookup decides what to share, and CONTINUUM decides how to expose the result as a contiguous tensor to unmodified kernels.
>
> **3. Agentic deployments / disaggregated prefill-decode.**
> Thank you for asking about these emerging serving settings. We agree that this is an important direction to clarify.
>
> For agentic workloads involving branching or frequent context reuse, this is precisely the type of topological dynamism CONTINUUM targets. However, we should be clear that the current implementation is single-GPU (as discussed in the Future Work appendix). Thus, for cross-device prefill/decode disaggregation, additional support for cross-device translation/transfer would be needed. We will revise the text to distinguish what is already supported today (single-device branching/prefix reuse) from what remains in future work (multi-GPU / disaggregated settings).

---

### Decision · Program_Chairs · 2026-04-30

**Decision:**

Accept (spotlight)

**Comment:**

The paper addresses an important problem and makes a strong practical contribution by making GPU VMM remapping usable for latency-sensitive dynamic inference while preserving contiguous tensor semantics. The main concerns were about novelty relative to prior VMM-based work, portability and implementation boundaries, and the evaluation under different branching settings. While these concerns were substantive, the rebuttal addressed them well. Therefore, I recommend acceptance and encourage the authors to incorporate the key rebuttal clarifications into the final version, especially regarding related work, limitations and portability, implementation details, and the beam/ToT-style evaluation settings.